# Effect of a School-Based Anxiety Prevention Program among Primary School Children

**DOI:** 10.3390/ijerph16244913

**Published:** 2019-12-05

**Authors:** Siti Fatimah Ab Ghaffar, Sherina Mohd Sidik, Normala Ibrahim, Hamidin Awang, Lekhraj Rampal Gyanchand Rampal

**Affiliations:** 1Faculty of Hospitality, Tourism and Wellness, City Campus, Pengkalan Chepa, Locked Bag 36, Kota Bharu 16100, Kelantan, Malaysia; fatimah.g@umk.edu.my; 2Department of Community Health, Faculty of Medicine and Health Sciences, Universiti Putra Malaysia, Serdang Selangor 43300, Malaysia; dr_rampal1@hotmail.com; 3Department of Psychiartry, Faculty of Medicine and Health Sciences, Universiti Putra Malaysia, Serdang Selangor 43400, Malaysia; normala_ib@upm.edu.my (N.I.); hamidin@upm.edu.my (H.A.); 4Cancer Resource & Education Center, Faculty of Medicine and Health Sciences Universiti Putra Malaysia, Serdang Selangor 43400, Malaysia

**Keywords:** anxiety, primary school, children, Malaysia

## Abstract

Anxiety is one of the most common mental health disorders in childhood, and children with anxiety have an increased risk of psychiatric disorders during adulthood. This study aimed to evaluate the effectiveness of a school-based anxiety prevention program for reducing anxiety among primary school students relative to a school-as-usual control group. Secondary to this, the current study aimed to examine the effect of a school-based prevention program on worry coping skills and self-esteem. A two-group parallel cluster randomized controlled trial of a single-blinded study was conducted to evaluate the effectiveness of the program, with schools as the unit of allocation and individual participants as the unit of analysis. The intervention program was conducted between May 2016 and December 2017. The primary outcome was anxiety, whereas the secondary outcomes were worry coping skills and self-esteem measured at three months post-intervention. Data were analyzed by using a generalized linear mixed model, accounting for the clustering effect. Subgroup analyses were performed for children with anxiety. A total of 461 students participated in this study. At baseline, there was no significant difference between groups for anxiety score, worry coping skills score, and self-esteem score (*p* > 0.05). The intervention was effective in reducing anxiety for the whole sample (*p* = 0.001) and the anxiety subgroup (*p* = 0.001). However, it was not effective in improving worry coping skills and self-esteem. These findings suggest that the program could be effective for reducing symptoms of anxiety when delivered in schools and provide some support for delivering this type of program in primary school settings.

## 1. Introduction

Anxiety disorders are one of the most common mental health disorders in both children [1] and adolescents [2]. Evidence suggests that children suffering from anxiety have an increased risk of psychiatric disorders during childhood or adulthood [3,4,5]. In their work, Bittner et al. [3] documented anxiety in children to be a predictor of a range of mental health problems in adolescence, such as panic attacks, depression, conduct disorder, and attention deficit/hyperactivity disorder. In a longitudinal study, researchers showed that adult major depression disorder was predicted by childhood anxiety [4]. 

Previous studies also showed that anxiety was significantly associated with suicide attempts [6] and suicidal ideation [7]. In a longitudinal study, Bolton et al. [6] reported that the presence of one or more anxiety disorders at baseline was significantly associated with subsequent-onset suicide attempts. Similarly, a study by Wolk et al. [8] stated that reduced childhood anxiety symptoms had a long-term effect in preventing suicidality in children.

Although effective treatments for childhood anxiety disorders are available [9], many children have inadequate accessibility to treatment [10] and very few children receive treatment for anxiety [11]. Both parental stigmatization [12] and treatment cost [13] are barriers to mental health service utilization among children. Barriers to mental health service utilization in a clinical setting, such as transportation and cost, could be minimized by conducting school-based anxiety prevention programs [14].

School-based anxiety prevention programmes could be implemented by targeting (i) all students regardless of symptom level (universal program), (ii) children who are at risk of developing anxiety disorder (selective program), or (iii) children with mild or early symptoms of anxiety (indicated program) [15]. However, universal programs reduce time as the screening of participants is not necessary [16], and they enable all children to participate in the program without the social stigma of “being selected” [17]. 

Findings from a systematic review show that interventions based on theory were more effective as compared to non-theory-based interventions [18]. Theories were derived from two main sources: stimulus response (SR) theory [19] and cognitive theory [20]. In stimulus response theory, the frequency of behavior is determined by its consequences or reinforcement, and the mere temporal relationship between a behavior and an immediately following reward is sufficient to increase the probability that the behavior is repeated [21]. Hence, reasoning or thinking concepts are not necessary to explain behavior. In contrast, mental processes are crucial elements of all cognitive theories. In cognitive theories, also known as value-expectancy theories, behavior is a function of the subjective value of an outcome and the subjective probability—the expectation—that that action will achieve that outcome.

Data from a systematic review also showed that childhood anxiety prevention programs are often based on cognitive behavioral therapy (CBT) [22]. CBT requires an ability to systematically identify, challenge, and generate alternative ways of thinking, which involves a degree of cognitive maturity and also requires an ability to engage in abstract tasks [23]. However, according to Piaget’s Cognitive Development Theory, children aged seven to twelve years of age are at the concrete operational stage, which is the beginning of logical or operational thought where children still struggle with abstract ideas [24].

Moreover, most CBT-based interventions focus on behavior and thoughts. Children with mental health problems had emotion-related gaps which may not be sufficiently prevented by targeting behavior and thought [25]. CBT-based interventions were found to insufficiently address the emotion-related discrepancies among children with anxiety [26]. In contrast, initial evidence for emotion-focused cognitive behavioral treatment was found to be effective in reducing childhood anxiety symptoms by Suveg and his colleagues [27]. However, a more recent study of an emotion-focused cognitive–behavioral program demonstrated that there was no significant decrease in anxiety symptoms between the intervention and control groups [28]. Similarly, results from a longitudinal study showed that there was no intervention effect between groups at 42- and 54-month follow-ups [29]. These non-significant findings could be due to the theory related to cognitive development, specifically that the cognitive component, which may require the ability for abstract thinking styles, may be too complex for children under twelve years old. Thus, anxiety prevention programs that contain more content on emotional competency skills and less complex cognitive components are needed.

Alternatively, an SR-based prevention program could be implemented to reduce childhood anxiety. Previous studies have used information–motivation–behavioral (IMB) theory [30], which is one example of SR theory, for preventing anxiety in the community [31]. Few published studies have addressed the possible effects of an IMB-based prevention program in reducing childhood anxiety symptoms. A school-based anxiety prevention program was designed in this study for primary school children who are at the concrete operations cognitive development level, at which the mental processes which are required for cognitive theories are not sufficiently mature. Hence, the intended intervention was developed based on information—motivation—behavioral skills (IMB) model. The current study aimed to evaluate the effectiveness of an IMB-based anxiety prevention program among primary school children.

## 2. Materials and Methods 

### 2.1. Study Design and Participants

This study involved a two-group parallel cluster randomized controlled trial in children attending government primary schools, with the school as the unit of allocation and the individual as the unit of analysis (see Figure 1). The study was undertaken in Jerantut, Pahang, Malaysia, between May 2016 and December 2017. Sampling with probability proportionate to size [32] was used as the sampling method. Twelve government-funded primary schools in a district on the east coast of Peninsular Malaysia were invited to participate in this study through an invitation letter and a brief explanation of the study. Children who could not comprehend and write in English or the Malay language or who were non-citizens were excluded. All children aged 10 and 11 years (Standard 4 and Standard 5) in the participating schools were eligible. The allocated intervention was given to all participants in the intervention groups universally as part of the extra co-curriculum classes.

The sample size for this study was calculated by using comparison of means for a cluster randomization design. The intracluster coefficient (ICC) was 0.01 and the cluster size was 55 (mean cluster size). The minimum sample size needed for this study was 105. After adjustment for non-response rate, attrition rate, and design effect, the sample size needed was 630. Therefore, 12 schools were selected to participate in this study. 

### 2.2. Randomization and Masking

Once all the schools were enrolled, we randomly assigned schools (1:1) to be an intervention or control school. Randomization was conducted at the school level in order to minimize possible contamination within schools. A simple randomization protocol was used to randomize schools into the control and intervention groups. Firstly, numbers were assigned to the schools. The numbers were written on pieces of paper, folded, and mixed up. The numbers were then picked at random. The first six primary schools were assigned as intervention schools and the remaining schools were assigned as control schools. 

Children were masked to the intervention allocation to maintain the blinding process throughout the study. The consent and information sheets for respondents and their parents or guardians only informed about the program in general; they did not contain specific information about the program they were going to receive—either intervention or control. 

### 2.3. Intervention

The intervention consisted of six modules. The first module contained an introduction to anxiety. The main components of the second module were types of emotions and the range of emotion intensity. The focus of module three was emotion triggers and the effect of emotions on the body. Empathy skills were covered in module four and emotion regulation skills were the main components in module five. The main objective of module six was self-esteem in children. The intervention trialed in this study consisted of four 60 min weekly sessions delivered to whole classes of children.

### 2.4. Procedures

The intervention program was delivered in schools and provided to whole classes of children. Children had their own worksheet, and research assistants had a detailed session plan that specified key learning points, objectives, and core activities for each session. 

In the intervention groups, each session was led by two trained research assistants. All research assistants had at least an undergraduate university degree. A half-day of training was given to the research assistants in order to train them on the program modules and their role in every session. They were given a program manual for guidance. They were allowed to ask any questions about the modules and for clarification before conducting the program. Ongoing supervision was provided by the program developer.

In the control group, children participated in the usual extra co-curriculum classes provided by the school. All schools were following a Malaysia National Curriculum Programme. The sessions were planned and provided solely by the teacher and did not include any external input from the research team. Students in the control schools were offered the same program after the study was completed.

### 2.5. Measures

Child outcomes were collected during class time with self-completed questionnaires administered by research assistants at baseline, post-intervention (immediately after program completion) and three months post-intervention. The primary outcome was symptoms of anxiety three months post-intervention as established by the short version of the Revised Child Anxiety and Depression Scale (RCADS 25) [33]. Secondary outcomes were measured using the Child Worry Management Scale (CWMS) to assess worry [34] and the Rosenberg Self-Esteem Scale for self-esteem [35]. 

In the RCADS 25, 15 items assessed anxiety and 10 items assessed depression. Respondents were asked to indicate how often each item applied to them according to a 4-point Likert scale (0 = never, 1 = sometimes, 2 = often, or 3 = always). The total score for each item included in the subscale was used to compute a score. A T score of 65 or higher indicated the borderline clinical threshold, whereas a T score of 70 or higher indicated a child above the clinical threshold. A T score below 65 indicated a normal case. 

The CWMS consisted of three subscales: (i) inhibition (the suppression of worry), (ii) dysregulation (exaggerated display of worry), and (iii) coping (constructive ways of managing worry). All items were answered by using a 3-point Likert scale. Responses were added for each item included in the subscale to compute a summary subscale score.

The RSES consisted of a 10-item scale that determines global self-worth by measuring both positive and negative feelings about the self. All items were answered using a 5-point Likert scale format ranging from strongly agree to strongly disagree. The scoring method for Rosenberg’s Self-Esteem Scale (RSES) is based on the total scores for all 10 items. Scores remained on a continuous scale. Higher scores indicate higher self-esteem.

### 2.6. Data Analysis

Descriptive statistics were used to assess group differences at baseline. One-way ANOVA was used for continuous data, whereas chi square test/Fisher’s exact test were used for categorical data. At baseline, these tests were performed for primary (anxiety) and secondary outcomes (worry coping and self-esteem) to ensure that both the intervention and control groups were comparable prior to implementation of the intervention program. 

We used generalized linear mixed model (GLMM) analysis to examine the effect of the intervention on primary (anxiety) and secondary outcomes (worry coping skill and self-esteem). The baseline data were adjusted in GLMM analysis. In this analysis, the clustering effect (school-level effects) was also adjusted. The 95% confidence interval (95% CI) was set for means estimation, with a *p*-value at 0.05 indicating the level of significance for rejection of the null hypothesis. 

Gender and depression were included in this analysis as covariates. The models comprised group interaction as well as group-by-time interactions using a first-order autoregressive structure. The analysis was suitable for repeated measures of a single subject. 

With regards to the model fit, the information criterion based on 2-log-likelihood (Akaike corrected and Bayesian) was used as an indicator. Smaller information criterion values indicate that the model fits better. Comparisons between the intervention and control groups at all three time points—baseline, immediately post-intervention, and three months post-intervention—were performed. For anxiety, a subgroup analysis was run to assess the effect of intervention among children with anxiety.

### 2.7. Ethical Approval

The study was approved by the Ethics Committee for Research Involving Human Subjects, Universiti Putra Malaysia. Participation required written consent from the school headmasters and from parents not opting their child out of the study, in addition to signed assent from the child.

## 3. Results

### 3.1. Participant Characteristics

Table 1 shows the participant characteristics relating to socio-demographic (age, gender, and ethnicity) and psychological variables (anxiety, worry coping, and self-esteem) at baseline. At baseline, there were no significant differences between the intervention and control groups for both categorical and continuous variables (*p* > 0.05).

### 3.2. Intervention Effects

Table 2 shows the mean anxiety, worry coping skills, and self-esteem scores at all three time points for the intervention and control groups. For the primary outcome, we found a statistically significant difference between the intervention and control groups. The results indicate that there was a significant difference in anxiety scores between children who received the intervention and children allocated to the control, with those in the intervention group reporting lower anxiety scores (F(4,1097) = 5.856, *p* = 0.001) (see Table 3). 

We also conducted separate subgroup analysis in the children with anxiety. We noted between-group differences in the mean RCADS at three months post-intervention (see Table 4). The results demonstrated that significantly more children in the intervention group than in the control group had a reduced anxiety score (F(4,53) = 6.760, *p* = 0.001).

## 4. Discussion

The findings from this study indicate that our school-based anxiety prevention program was effective in reducing anxiety scores. The reductions in anxiety scores were significantly greater for participants in the intervention group; however, this difference was small. This is a positive result given that the intervention is a relatively short and inexpensive four-week program without any additional or booster sessions. Thus, the hypothesis that the intervention group showed a greater reduction in anxiety scores compared to the control group was supported. 

The findings of our study are supported by previous studies on school-based anxiety prevention programs for primary school children [36,37,38,39]. The differences between anxiety scores in the intervention and control groups were larger in previous studies compared to in our study. This could be due to longer periods of follow-up, specifically 12 months post-intervention compared to three months post-intervention in the current study, which may have allowed participants in the intervention group to practice skills they learned during this period. 

The findings of our study also corresponded to the findings of a universal school-based anxiety prevention study among children from socio-economically disadvantaged communities in Australia, where anxiety significantly reduced over time from baseline to 12 months post-intervention [39]. However, a limitation of this study was the absence of a control group. Thus, placebo and maturation effects cannot be conclusively discounted. 

In contrast, a universal school-based anxiety prevention program conducted in Australia showed inconsistent findings with this study, where there was no significant reduction in anxiety scores between the intervention and control groups at 30 months post-intervention [28]. The long-term effects were also measured, and no significant reduction in anxiety score between the groups was obtained at 42- and 54-month follow-ups [29]. This may be due to the content of the intervention modules and the method of delivery. Specifically, the cognitive component of the module may have been too complex for the children, and the program was delivered to the whole classroom as opposed to the method used in the current study, in which we delivered the program to smaller groups. 

The findings from a cluster-randomized trial conducted in Canada were also different to the findings from our study. The results from this study revealed that the anxiety prevention program did not effectively reduce anxiety [40]. Miller and his colleagues conducted the study among Aboriginal children, where socio-demographic and cultural differences could have contributed to the contradictory findings of their study [40]. Similarly, no intervention effect between groups was reported in a study by Manassis et al. [41]. This dissimilarity of findings could be due to different types of intervention programs and the lack of a non-intervention comparison group. A quasi-experimental study evaluating the effectiveness of an anxiety prevention program for children in Japan also revealed diverse findings [42]. This could be due to small sample size, no randomization process, and different recruitment methods and questionnaire measures compared to the current study. 

The findings from the current study showed that the effect size of our school-based anxiety prevention program at three months post-intervention was small, with Cohen’s *d* = 0.10 [43]. This is consistent with meta-analytic reviews on anxiety prevention programs in children and adolescents. Here, a small effect size on anxiety was found at the short-term follow-up [44,45,46]. A systematic review and meta-analysis focused on school-based anxiety and depression prevention programs in young people was conducted to provide a comprehensive overview on intervention effects at long-term follow-up. Small effects were found at a 12-month follow-up for anxiety [47]. 

Findings from the systematic review revealed that the effect size ranged from small to moderate for universal school-based anxiety prevention programs which have a primary focus on anxiety prevention, whereas universal school-based anxiety prevention programs that have a dual focus on anxiety and depression prevention demonstrated non-significant findings at immediate and long-term follow-ups [48]. Our universal school-based anxiety prevention program has a primary focus on anxiety prevention and was effective in reducing anxiety with small effect size at the short-term follow-up (three months post-intervention). 

In this study, worry coping skill was our secondary outcome. There was no significant difference in worry coping skill scores between the intervention and control groups at three months post-intervention. The findings of this study are in line with those of a clustered randomized controlled trial, where no group differences between groups’ worry coping skill scores were observed between the intervention and control groups [38]. One possible reason for the non-significant finding in our study could be due to the short length of the intervention program and follow-up. The application of knowledge and skills in practice may take a longer period of time. Measurement at three months post-test may be too early to detect changes in the children’s worry coping skill. Future research should include a longer follow-up time point and observe if this has an impact on worry coping skills. 

In contrast, Bouchard and his colleagues [49] had different findings from our study. Their primary anxiety prevention program for children significantly improved children’s coping skills. The program was delivered using story books to describe characters facing common stressors and included strategies to manage day-to-day difficulties. The findings from our study were also inconsistent with a recent randomized controlled trial that evaluated the effectiveness of an emotion-focused cognitive behavioral therapy anxiety prevention program for children with anxiety. A total of 92 children aged 7–12 years were involved and randomly assigned to emotion-focused or traditional CBT anxiety prevention programs. The results showed that emotion regulation among children from both groups was improved [50]. The non-significant findings from the current study may be due to the lack of additional sessions for them to boost their worry coping skills.

The other secondary outcome measured in this study was self-esteem. The hypothesis that the self-esteem of children allocated to receive the intervention would increase compared to that of the children in the control group was not supported. Self-esteem scores decreased from baseline to three months post-intervention in both groups. There was no significant intervention effect between groups over time. These findings are in line with a previous study where no effects between groups on self-esteem scores were found [38]. The decrease in self-esteem scores could be due to other environmental factors; for example, the school mid-term examination was conducted between baseline and three months post-intervention in this study.

In the anxiety subgroup, there was a group-by-time interaction with regards to anxiety score. The anxiety scores for the intervention group were significantly reduced from baseline to three months post-intervention as compared to the control group. Stallard and his colleagues [38] found comparable findings to those from our study, where an intervention effect between groups was found within children with anxiety. 

The strength in this study was the usage of the national language among the local population, which was the Malay language. The use of this language for each questionnaire was pretested and validated. The other strength of this study was the study design. It was a cluster randomized controlled trial; random assignment was carried out at school level to minimize cross contamination between the intervention and wait-list control groups. The response rate was high with an adequate sample size, and data were analyzed by using appropriate statistical analysis (the clustering effect was taken into consideration). 

There were also several limitations found in this study. Firstly, self-reported questionnaires were used in this study. This could lead to recall bias and reporting bias. Therefore, in future research, parents’ and teachers’ perspectives could be taken into consideration. Secondly, the specific population of this study (children attending government primary schools in rural areas) was not representative of all children from other areas, such as urban areas. Overall, the findings of this study might not apply to the general population. Exclusion of non-Malaysians and the short-term follow-up (three-month follow-up) were also limitations of this study.

Future studies could provide booster sessions or additional sessions for the participants at six months post-intervention to allow the children to practice the skills learned. Story books as an interactive intervention strategy could also be integrated into future studies to reduce anxiety in primary school children. Future research could measure the effectiveness of the school-based anxiety prevention program over longer periods of time, for example, at 6 months and 12 months post-intervention, to determine the sustainability of the program. The involvement of school teachers could also be considered in future studies because children spent a lot of their effective time in school. Teachers act as role models in practicing emotion coping skills and dealing with negative emotions. 

## 5. Conclusions

The current study presents important findings on the delivery of an anxiety prevention program targeting primary school children within the school setting. The school-based anxiety prevention program was effective in reducing anxiety scores and could be implemented in the local school setting by integrating it into the school curriculum, thus increasing children’s access to mental health care.

## Figures and Tables

**Figure 1 ijerph-16-04913-f001:**
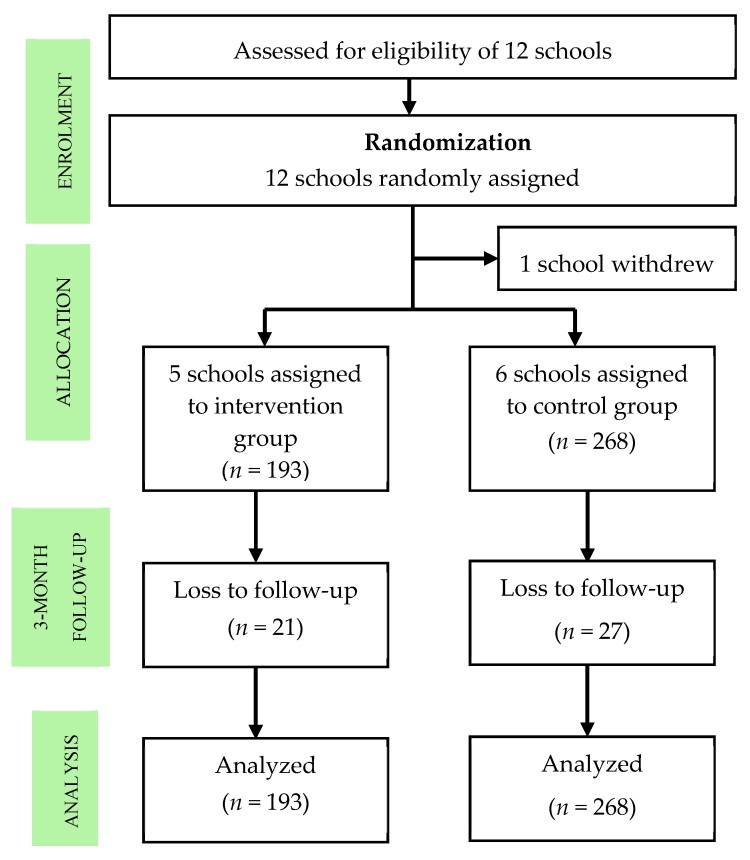
CONSORT (Consolidated Standards of Reporting Trials) diagram of the clusters and study participants.

**Table 1 ijerph-16-04913-t001:** Baseline characteristics relating to the socio-demographic and psychological variables of the participants.

Variables	Intervention (*n* = 193)	Control (*n* = 268)	*p*-Value
**Age: *n* (%)**			
10	98 (50.8)	135 (50.4)	0.932 ^a^
11	95 (49.2)	133 (49.6)	
**Gender: *n* (%)**			
Male	77 (39.9)	126 (47.0)	0.129 ^a^
Female	116 (60.1)	142 (53.0)	
**Ethnicity: *n* (%)**			
Malay	192 (99.5)	268 (100.0)	0.238 ^a^
Non-Malay	1 (0.5)	0 (0.0)	
**Anxiety score: Mean (SD)**	14.14 (6.194)	15.06 (6.948)	0.156 ^b^
**Worry coping skills score: Mean (SD)**	5.35 (1.409)	5.54 (1.518)	0.174 ^b^
**Self-esteem score: Mean (SD)**	37.43 (5.724)	36.46 (6.127)	0.089 ^b^

SD—Standard deviation. ^a^ Chi-Square/Fisher exact test; ^b^
*t*-test.

**Table 2 ijerph-16-04913-t002:** Mean anxiety, worry coping skills, and self-esteem scores at all three time points for the intervention and control groups.

Outcome Variable	Intervention			Control		
	Baseline Mean (SD)	Immediately after Intervention Mean (SD)	3 Months after Intervention Mean (SD)	Baseline Mean (SD)	Immediately after Intervention Mean (SD)	3 Months after Intervention Mean (SD)
Anxiety score	14.14 (6.194)	14.10 (6.421)	12.95 (7.088)	15.06 (6.948)	15.14 (6.325)	13.87 (7.178)
Worry coping skills score	5.35 (1.409)	5.28 (1.434)	5.26 (1.382)	5.54 (1.518)	5.58 (1.487)	5.50 (1.443)
Self-esteem score	37.43 (5.724)	36.41 (6.638)	36.71 (7.106)	36.46 (6.127)	36.68 (6.710)	35.52 (6.834)

**Table 3 ijerph-16-04913-t003:** Effect of the school-based anxiety prevention program on anxiety, worry coping skills, and self-esteem scores for all participants.

Outcome Variable	Parameter	F	df1	df2	*p*-Value	Effect Size (Cohen’s *d*)
Anxiety score	Group	1.213	1	1097	0.271	
	Group × time	5.856	4	1097	0.001 *	0.1026
Worry coping skills score	Group	2.563	1	1098	0.110	
	Group × time	0.484	4	1098	0.748	
Self-esteem score	Group	0.058	1	1075	0.810	
	Group × time	1.914	4	1075	0.106	

* *p* < 0.001.

**Table 4 ijerph-16-04913-t004:** Effect of the school-based anxiety prevention program on anxiety for participants with anxiety.

Outcome Variable	Parameter	F	df1	df2	*p*-Value	Effect Size (Cohen’s *d*)
Anxiety score	Group	0.030	1	53	0.898	
	Group × time	6.760	4	53	0.001 *	0.3999

* *p* < 0.001.

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
