# Peer review of "Effect of a School-Based Anxiety Prevention Program among Primary School Children"

_ijerph, 2019, doi:10.3390/ijerph16244913_

Round 1

Reviewer 1 Report

General comments

=============

This paper reports on the effectiveness of a school-based anxiety prevention program among primary school students. Schools play an important role in supporting young people’s mental health as the prevalence of anxiety and depressive disorders among young people is high. There is a real need for prevention as many young people experiencing mental health symptoms are at increased risk of further disease progression, functional impairment and suicidality. School-based programs are a great way to provide early intervention whilst reducing costs and time and overcoming common barriers such as access and stigma. Overall, the study has merit, however the manuscript does need some work to make the background argument and rationale for the study clearer and the discussion clearer and more concise. Some recommended revisions are noted below.

Specific comments

============

ABSTRACT:

Suggestion to insert a sentence that includes a bit of background, i.e., Why is it important to examine the effect of a school-based anxiety program. Suggestion to reword the first sentence to: “This study aimed to evaluate the effectiveness of a school-based anxiety prevention program for reducing anxiety among primary school students relative to a school-as-usual control group. Secondary to this, the current study aimed to examine the effect of school-based prevention program on worry coping skills and self-esteem”. The abstract could include the length of the study period and information about how many students participated in the study. Line 28: Replace “skill” with “skills” Line 29: Suggestion to replace recommend with “suggest” or change the sentence to: “These findings suggest that the program could be effective for reducing symptoms of anxiety when delivered in schools and provide some support for delivering this type of program in primary school settings".

INTRODUCTION:

Line 34: Remove “found to be”; remove the comma after both Line 35: remove “were” Line 37: Remove “are found” Line 38: Replace “for example” with “such as” Line 44: Suggestion to reword sentence to: “Similarly, a study by Wolk et al (2015) reported that a reduction of anxiety symptoms in childhood confers long-term protection from suicidality in children” to avoid the repetition of the phrase “longitudinal study”. Line 46: Remove comma after “although” Line 46: Reword sentence: “Although effective treatments for childhood anxiety disorders are available [9], many children have limited access to treatment [10] and very few children with anxiety disorders receive treatment [11]. Line 49: Reword sentence: “These barriers and other common barriers to accessing mental health services such as transportation and cost could be minimised by implementing anxiety prevention programs in the school environment”. Line 55: Replace “reduces” with “reduce” Line 58: Replace start of sentence with: “Findings from a systematic review show that interventions based on theory were found to be …” Line 61: Insert comma after “reinforcement” The paragraph (line 58 to 64) could be fleshed out a little more. Line 65: Insert the word “a” after “Data from” Line 70: Remove the word “the” before logical, replace “though” with “thought” and replace “and still struggle with abstract ideas” with “where children still struggle with abstract ideas” Line 71: Replace “focused” with “focus” and “thought” with “thoughts” Line 72: Reword sentence to: “…which may not be sufficiently prevented by targeting behaviour and thought [25]”. Line 73: Replace “not completely” with “to insufficiently” Line 74: Insert “In contrast” before “initial evidence” Line 76: Remove the word “in” after However, Line 78: Reword sentence to: “Similarly, results from a longitudinal study showed that there…” Line 80: Replace “The” with “These non-significant findings…” insert “specifically” after “development” and before “the cognitive…” Line 82: Replace “style” with “styles” Line 82: Reword sentence: “Thus, anxiety prevention programs that contain more content on emotional competency skills and less complex cognitive components are needed”. Line 84: Replace “study” with “studies” and insert “have used” Line 86: Remove “of”, replace “examples” with “example”, replace “theories” with “theory”. Line 88: Replace “we therefore aimed” with “The current study aims”. Very little attention is given to "IMB theory" which seems to have been discussed very briefly at the end. Given that the study aims to evaluate the effectiveness of a prevention program based on IMB, this section needs to be integrated with the previous section more and fleshed out a bit. 

MATERIALS AND METHODS:

Line 92: Replace “We did” with “A” two-group… Line 94: Insert “the” before “sampling method”. Line 96: Insert “an” before “invitation letter”, insert “a” after “and”, replace “on” with “of”. Line 97-98: Reword sentence: “Children who could not comprehend and write in English or Malay language or non-citizens were excluded. Can the authors please clarify if the intervention was given to all participants in the intervention group. Some more information about when the study took place, for example., “The study was undertaken in XX, Malaysia, between February and June 2018”. Did the authors have a recruitment target? Why were only 12 school selected? The CONSORT diagram says “Assessed for eligibility of 12 schools” – can the authors please clarify what this means as the methods only describes inclusion and exclusion criteria for students and not schools. Were these schools co-ed, single sex, were they similar in size? The randomisation procedure is not ideal. Best practice would be to use excel or a random number generator to randomise schools rather than pulling them out of a hat. Line 119: insert “the” between at school. Line 120: Insert “A” before simple randomisation Line 121: replace “two groups” with “the” Line 122: replace “in” with “on”, remove “then”, start sentence with “The numbers” Line 123: Replace “another” with “the remaining” Line 125: Remove “With regards to the blinding procedure” and just start sentence with “Children were masked to the intervention allocation to maintain the blinding process throughout the study”. Line 127: Replace “mention about” with “contain specific information about”, insert “going” between “were” and “to” Line 128: Remove last sentence. Line 131: Replace “units” with “modules” Restructure paragraph to read: “The first module contained an introduction to anxiety. The main components of the second module were types of emotions and the range of emotion intensity. The focus of module three was emotion triggers and the effect of emotions on body. Empathy skills were covered in module four and emotion regulation skills were the main components in module five. The main objective of module six was self-esteem in children.” Line 137: Start sentence with “The intervention…” Suggestion to move the second sentence into the “intervention” section as it describes the intervention and not the procedure. Line 143: replace “assistant” with “assistants” Line 144: replace “module” with “modules” Line 145: replace “as a” with “for”, replace “module” with “modules” and insert “for clarification before conducting the program”. Line 148: Replace “groups” with “group” Line 153: Suggestion to reword sentence to: “baseline, post-intervention (immediately after program completion) and 3 months…” Line 157: Remove “assessed by” and replace with “were measured using the Child Worry Management Scale to assess worry [34] and self-esteem measured by the Rosenberg Self-Esteem Scale [35].” Line 160: Start sentence with “Descriptive statistics were used to assess group differences at baseline.” Line 162: Remove “the” after “At” Line 163: Replace “outcome” with “outcomes” and remove “outcomes” before “to ensure”. Replace “groups” with “the” Line 166: Insert “on” between “intervention” and “primary” Line 170: Replace “was” with “were” and remove “that may affect the outcome”. Line 172: Remove “both” Line 174: Replace “Value” with “values” and “indicated” with “indicates” Line 176: Insert a comma after anxiety, Line 177: Remove the first “subgroup” Further information about the questionnaires used such as cut off scores etc would be useful.

RESULTS

Line 186: Remove: Whereas” – just begin sentence with “Table 2” Line 187: Remove “the” Line 203-205: Reword paragraph to read: “For the primary outcome, we found a statistically significant difference between the intervention and control groups. Results indicate that there was a significant difference in anxiety scores between children who received the intervention compared to children allocated to control, with those in the intervention group reporting lower anxiety scores (F (4, 1097) = 5.856, p = 0.001) (See Table 3). Line 206: Replace “skill” with “skills” and “score” with “scores” in table title It would be good to include a table with the mean anxiety, worry and self-esteem scores at all three time-points for those in the intervention and control groups Line 209: Replace “did” with “conducted” Line 210: Replace “Result” with “Results” With the subgroup analysis in children with anxiety, what cut-off was used?

DISCUSSION:

The discussion needs some work to reduce the length of it, make it more concise and clearer. Line 216: replace “indicated” with “indicates” Line 217: Replace “of” with “in” Line 218: Consider rewording sentence to read: “The reductions in anxiety scores were significantly greater for participants in the intervention group, however, this difference was small. This is a positive result given the intervention is a relatively short and inexpensive four-week program without any additional or booster sessions”. The authors go on to describe numerous studies that do and do not support their results and why this may and may not be the case. I suggest that they try and discuss all of the studies that support their results in one (or a few more) paragraph and spend another paragraph (or a few more) discussing results that are different to their results. Too much detail is included about numerous other studies which is not necessary. They can they use the discussion to discuss other aspects of the study and what the results mean in a broader sense. Line 224: Replace “was” with “are”, remove “other” Line 225-226: Reword sentence: “The differences between anxiety scores in the intervention and control groups was larger in previous studies compared to our This could be due to longer periods of follow-up, specifically 12-months post-intervention compared to three-month post-intervention in the current study which may have allowed participants in the intervention group to practice skills they learned during this period.” Line 230: Replace “finding” with “findings” Line 232: Remove statistics (F and p values) Line 233: Suggestion to reword sentence to: “However, a limitation of this study was the absence of a control group. Line 239: Replace “where there was” with “and no significant…” Line 240: Suggestion to reword the sentence to: “This may be due to the content of the intervention modules and the method of delivery. Specifically, the cognitive component of the module may have been too complex for the children and the program was delivered to the whole classroom as opposed to the method used in the current study which delivered the program to smaller groups”. Line 245: Replace “Their” with “Results from this study revealed that…” Line 250: Replace “with” with “to”, reword sentence to: “Anxiety was found reduce significantly over time; however, no intervention effect between groups was reported [41]”. Line 255: replace “there was possibility” with “it was possible” Line 257: Consider rewording to: “A quasi-experimental study evaluating the effectiveness of an anxiety prevention program for children in Japan also revealed diverse findings. Differences in anxiety scores between groups approached significance at three-month follow-up [42]. The study recruited a small sample of thirteen children aged nine to 12 years in the intervention and sixteen children in the control group. The study did not randomize the children to the intervention and control conditions and used different recruitment methods to recruit children to each group. In addition, anxiety was measured using a different questionnaire measure compared to the current study”. Line 269-276: With the exception of the first sentence (which I would move to the start of the next paragraph), I would remove this whole paragraph from the discussion. This information does not belong in the discussion as does not add anything. Line 277: Start this paragraph with: “Findings from the current study showed that the effect size of our school-based anxiety prevention program at 3-month post-intervention was small (Cohen’s d = 0.10). This is consistent with a meta-analytic review on anxiety prevention programs in children and adolescents. Here, a small effect size (0.19) on anxiety was found at short-term follow-up. This finding is also in line with results from another meta-analytic review …”. Line 282: reword sentence to “However, these meta-analyses included both universal anxiety prevention programs and other types of prevention program, such as, targeted anxiety prevention programs [44, 45]. A more recent systematic review focused only on universal anxiety prevention programs in children revealed consistent findings with our study and reported a small effect size (0.11) at short-term follow-up [46]”. Line 289: Replace “setting was” with “settings is” Line 290: Replace “program” with “programs” Line 291: Replace “his” with “her” Line 292: Replace “effect” with “effects”, replace “was also” with “were found” Line 295: Replace “colleague” with “colleagues" Line 302: replace “program which has” with “programs which have” Line 303: Replace “program that has” with “programs that have a dual…” Line 307: Delete final sentence in this paragraph. Line 311: Replace “score” with “scores” Line 312: Suggestion to reword: “The findings of this study are in line with a clustered randomized controlled trial, where no group differences between groups worry coping skill scores were observed between the intervention and control groups [38]. One possible reason for the non-significant finding in our study could be due to the short length of the intervention program and follow-up. Application of knowledge and skills into practice may take a longer period of time. Measurement at three-month post-test may be too early to detect changes in children’s worry coping skill. Future research should include a longer follow-up time point and observe if this has an impact on worry coping skills. Line 319: Replace ”measured” with “measure” Line 321: Replace “also found an almost similar finding with our study,” with “reported similar findings to the current study” Line 323: remove “they” Line 329: Remove “by” and insert “included” after and at the end of the line Line 330: Replace “was” with “were” Line 331: Replace “evaluate” with “evaluated” Line 334: Replace “Finding from Suveg et al’s study showed” with “Results showed” Line 335: Reword sentence to: “The non-significant findings from the current study may be due to the lack of additional sessions for them to boost the worry coping skills” Line 337: Do the authors have a reference to support their statement: “There is a need for children to have at least one booster session for them to improve their worry coping in future studies”. What evidence are they basing this on? Line 339-344: Consider rewording to: “The other secondary outcome measured in this study was self-esteem. The hypothesis that self-esteem of children allocated to receive the intervention would increase compared to the children in the control group was not supported. Self-esteem scores decreased from baseline to 3-month post-intervention in both groups. There was no significant intervention effect between groups over time. These findings are in line with a previous study, where no effects between groups on self-esteem scores was found [38]”. Why do the authors think the self-esteem scores became worse from baseline to 3 months? Did the authors have ay suggestions about how the program could be improved? Is the program sustainable for future delivery? How was it received by schools and the school community, ie., was it acceptable?

CONCLUSION:

Consider rewording: “The current study presents important findings for the delivery of an anxiety prevention program targeting primary school children within the school setting. The school-based anxiety prevention program was effective in reducing anxiety scores and could be implemented in the local school setting by integrating it into the school curriculum, thus increasing access to mental health care to children.

Reviewer 2 Report

This article addresses the effectiveness of anxiety prevention program targeting children in primary schools using the experimental study design based on IMB theory what brings new view on this topic. I would like to highlight good methodology and procedure of the study, whole article clearly and comprehensibly describes the background, process of collecting data and its processing into particular outcomes. However, there are a few comments which need to be solved.

Descriptive Table 1 and 2 should be joined into one table with indicating N, %, mean(SD) and p value, what are the most important information from descriptive analysis, example how to join categorical and continuous variables in one table is in article (Descriptive table 1). Then, change also title of the table:

Kopcakova J., Dankulincova Veselska Z., Kalman M., van Dijk J.P., Reijneveld S.A.: Do motives to undertake physical activity relate to physical activity in adolescent boys and girls? International Journal of Environmental Research and Public Health, 2015, 12:7656-7666.

Discussion   

Need to be definitely condensed. As it is now, it is too long for readers. Avoid reporting particular numbers of effect sizes, p-numbers…. Some parts, eg. P8 L257-268, L277-L286,….. do not need to be described so in detail. Try to interpreted your results in connection with results of other studies in the way if they are supported or no and pay attention to discussion what does it mean and/or what new your study adds to existing knowledge. 

Round 2

Reviewer 1 Report

The manuscript is much improved. However, there are still some areas that need work.

Abstract:

Remove “Primary schools were randomly assigned to intervention and control schools. Students from intervention schools received school-based anxiety program while students from control schools received usual curriculum class.” This information is now captured in the preceding sentences.

Remove “which” from line 27

Remove “The level of significance was set at alpha = 0.05.”

Introduction:

Line 49 - Reword to: “Similarly, a study by Wolk et al [8] stated that reduced childhood anxiety symptoms had long-term effect in preventing suicidality in children”.

Line 51: Remove “however,”

Line 63: Reword to: Findings from a systematic review show that interventions based on theory were found to be more effective as compared to non-theory based interventions [18]

Line 82: Remove “to” after insufficiently

Line 149 – remove semi colon after the;

Lines 185-187: Reword to say: "Fifteen items assessed anxiety and 10 items assessed depression".

The specific details of each item for each subscale do not need to be included as this information is easily attainable with questionnaires and assessment measures.

Line 189 – Reword to: "The total score for each item included in the subscale was used to compute a score."

Line 195 – Reword to: Responses were added for each item included in the subscale to compute a summary subscale score.

Remove the following from lines 201-204: "Each item has its own score. “Strongly Disagree” was given 1 point, “Disagree” was given 2 points, “Neutral” was given 3 points, “Agree” was given 4 points, and “Strongly Agree” was given 5 points. Whereas, items 3, 5, 6 and 9 were reverse scored. Then, the total sum scores for all ten items were calculated."

This specific information about scoring is easily obtained and therefore not necessary to include.

Line 271 – replace “indicates” with “indicate”

Line 279 – insert “Study” after “our”

Line 285 – reword to: “where anxiety significantly reduced over time,…”

Line 340 – Replace with “findings”

Line 410 – Reword to: “…for example, at six-month and 12-month post-intervention…”

The discussion is still very very long and needs to be condensed. There is still far too much detail about individual studies and this could be summarised with out going into so much detail for each individual study. Perhaps the authors could read other manuscripts that also evaluate school-based programs and have a look at how their discussions are written and structured.
